# Leveraging Self-Paced Semi-Supervised Learning with Prior Knowledge for 3D Object Detection on a LiDAR-Camera System

Pei An [1], Junxiong Liang [2], Xing Hong [2], Siwen Quan [3], Tao Ma [4], Yanfei Chen [1], Liheng Wang [1] and Jie Ma [2,*]

[1] School of Electrical and Information Engineering, Wuhan Institute of Technology, Wuhan 430205, China
[2] School of Artificial Intelligence and Automation, Huazhong University of Science and Technology, Wuhan 430072, China
[3] School of Electronic and Control Engineering, Changan University, Xi'an 710064, China
[4] Institute of Computer Application, China Academy of Engineer Physics, Mianyang 621900, China
[*] Correspondence: majie@hust.edu.cn

**Abstract:** Three dimensional (3D) object detection with an optical camera and light detection and ranging (LiDAR) is an essential task in the field of mobile robot and autonomous driving. The current 3D object detection method is based on deep learning and is data-hungry. Recently, semi-supervised 3D object detection (SSOD-3D) has emerged as a technique to alleviate the shortage of labeled samples. However, it is still a challenging problem for SSOD-3D to learn 3D object detection from noisy pseudo labels. In this paper, to dynamically filter the unreliable pseudo labels, we first introduce a self-paced SSOD-3D method SPSL-3D. It exploits self-paced learning to automatically adjust the reliability weight of the pseudo label based on its 3D object detection loss. To evaluate the reliability of the pseudo label in accuracy, we present prior knowledge based SPSL-3D (named as PSPSL-3D) to enhance the SPSL-3D with the semantic and structure information provided by a LiDAR-camera system. Extensive experimental results in the public KITTI dataset demonstrate the efficiency of the proposed SPSL-3D and PSPSL-3D.

**Keywords:** 3D object detection; semi-supervised learning; self-paced learning; LiDAR-camera system

## 1. Introduction

Three dimensional (3D) environment perception has an important role in the field of autonomous driving [1]. It analyzes the real-time information of the surroundings to ensure traffic safety. To avoid vehicle collision [2], 3D object detection is an important approach among the techniques of 3D environment perception. Its task is to identify the classification and predict the 3D bounding box of a targeted object a the traffic scenario. In a word, 3D object detection performance affects the traffic safety of intelligent driving [3]. As 3D object detection requires spatial information from the environment, light detection and ranging (LiDAR) is a suitable sensor because it can generate a 3D point cloud in real-time [4]. Thanks to its ranging accuracy and stability, multi-beam mechanical LiDAR is the mainstream LiDAR sensor for environment perception [1,5]. It is referred to as LiDAR henceforth for discussion simplicity. Due to the limited rotation frequency and beam number, the vertical and horizontal resolution angles are limited, causing sparsity of the LiDAR point cloud, thus increasing the difficulty of 3D object detection [1].

The current 3D object detection method exploits the technique of deep learning and takes LiDAR points as the main input to identify and localize 3D objects [6]. To decrease the negative impact of the sparse LiDAR point cloud, researchers have done lots of work in several areas such as (i) detector architecture [7], (ii) supervised loss function [4], and (iii) data augmentation [8], which have made progress in fully supervised 3D object detection (FSOD-3D). By training with the sufficient labeled data, FSOD-3D can achieve performance in 3D environment perception close to that of humans.

However, there is a contradiction between the demand for 3D object detection performance and the cost of human annotation on the LiDAR point cloud. Due to the sparsity of the LiDAR point cloud and occlusion of the 3D object, the annotation cost of the 3D object is high, so the labeled dataset is insufficient. Therefore, it is essential to utilize unlabeled data to train the 3D object detector.

Semi-supervised 3D object detection (SSOD-3D) [9–11] has attracted a lot of attention, for it improves the generalization ability of the 3D object detector with both labeled and lots of unlabeled samples recorded in various traffic scenarios. From the viewpoint of optimization, SSOD-3D is regarded as a problem that alternatively optimizes the weights of 3D objects detector $\mathbf{w}$ and pseudo labels from the unlabeled dataset. For one unlabeled sample $\mathbf{x}$ (i.e., the LiDAR point cloud), its pseudo label $\mathbf{l}$ consists of the 3D bounding boxes of the targeted objects (i.e., car, pedestrian, cyclist), predicted from $\mathbf{x}$. This means that the capacity of $\mathbf{w}$ and quality of $\mathbf{x}$ are coupled. To obtain the optimal $\mathbf{w}^*$, it is essential to decrease the false-positives (FP) and true-negatives (TN) in $\mathbf{l}$. To improve the quality of $\mathbf{l}$, one common approach is to utilize the label filter to remove the incorrect objects in $\mathbf{l}$. Sohn et al. [12] employed a confidence-based filter to remove pseudo labels of which the classification confidence score is below threshold $\tau_{\text{cls}}$. Wang et al. [11] extended this filter [12] in their SSOD-3D architecture, with both the $\tau_{\text{cls}}$ and the 3D intersection-over-union (IoU) threshold. However, in practical application, the optimal thresholds are different with detector architecture, the training dataset and even the object category. It takes a lot of time to search the optimal thresholds in the label filter, which is inefficient in the actual application. Thus, it is a challenging problem to design a more effective and convenient SSOD-3D method.

In the background of intelligent driving, most self-driving cars are equipped with LiDAR and an optical camera. A sensor system with LiDAR and a camera is called a LiDAR-camera system. To remedy the sparsity of the LiDAR point cloud, researchers have studied 3D object detection methods on a LiDAR-camera system [13–17]; the LiDAR-camera system provides a dense texture feature from the RGB image, improving the classification accuracy and confidence of the 3D detection result. Thus, it is wise for the SSOD-3D to consider the prior knowledge provided by LiDAR-camera systems.

Motivated by this, we present a novel SSOD-3D method on a LiDAR-camera system. First, in order to train a 3D object detector with reliable pseudo labels, we introduce a self-paced, semi-supervised and learning-based 3D object detection (SPSL-3D) framework. It exploits the theory of self-paced learning (SPL) [18] to adaptively estimate the reliability weight of pseudo label with its 3D object detection loss. After that, we notice that the prior knowledge in the LiDAR point cloud and RGB image benefits the evaluation of the reliability of pseudo label, and propose a prior knowledge-based SPSL-3D (named PSPSL-3D) framework. Experiments are conducted in the autonomous driving dataset KITTI [19]. With the different labeled training samples, both comparison results and ablation studies demonstrate the efficiency of the SPSL-3D and PSPSL-3D frameworks. Therefore, SPSL-3D and PSPSL-3D benefit SSOD-3D on a LiDAR-camera system. The remainder of this paper is organized as follows. At first, the related works of FSOD-3D and SSOD-3D are illustrated in Section 2. In the next, the proposed SPSL-3D and PSPSL-3D methods are discussed in Section 3. After that, experimental configuration and results are analyzed in Section 4. Finally, this work is concluded in Section 5.

## 2. Related Works

### 2.1. Fully Supervised 3D Object Detection

To achieve high performance in environment perception in autonomous driving, FSOD-3D on LiDAR has been widely studied in recent years. Its architecture commonly has three modules [6]: (i) data representation, (ii) backbone network, (iii) detection head. LiDAR data mainly have three representations: point-based [4], pillar-based [20], and voxel-based [21]. Selection of a backbone network is dependent on data representation. Point-based features are extracted with PointNet [22], PointNet++ [23], or a graph neural network

(GNN) [24]. As the pillar feature is regarded as the pseudo image, a 2D convolutional neural network (CNN) can be used. To deal with the sparsity of the LiDAR voxel, a 3D sparse convolutional neural network (Spconv) [25] is exploited for feature extraction. The detection head can be classified as anchor-based [26] and anchor-free [27]. The anchor-based 3D detector first generates the 3D bounding boxes with the pre-defined size of the different categories (called anchors) that are placed uniformly in the ground, then predicts the size, position shift, and confidence score of each anchor, and removes the incorrect anchors with lower confidence scores. After that, to remove the redundant 3D bounding boxes, 3D detection results are obtained by using non maximum suppression (NMS) on the remaining shifted anchors. The anchor-free 3D detector first usually predicts the foreground point cloud from the raw points [28], and then predicts the 3D bounding box from each foreground LiDAR point with a fully connected (FC) layer. Then, NMS is exploited to remove the bounding boxes with high overlap.

Recently, many researchers have produced lots of work; Zheng et al. [29] trained a baseline 3D detector with knowledge distillation. The teacher detector generates the pseudo label, and supervises the student detector with shape-aware data augmentation. Schinagl et al. [30] analyzed the importance of each LiDAR point for 3D object detection by means of Monte Carlo sampling. Man et al. [31] noticed that LiDAR can obtain multiple return signals with a single laser pulse and use this mechanism to encode a meaningful feature for the localization and classification of 3D proposals. Yin et al. [27] proposed a light anchor-free 3D detector to regress the heat maps of 3D bounding box parameters. In the training stage, it did not need target object alignment, thus saving lots of time. Based on 3D Spconv [25], Chen et al. [32] presented a focal sparse convolution to dynamically select the receptive field of voxel features for convolution computation. This can be extended for the multi-sensor feature fusion. As for FSOD-3D on a LiDAR-camera system, Wu et al. [33] used multi-model-based depth completion to generate a dense colored point cloud of 3D proposals for accurate proposal refinement. Li et al. [34] presented the multi-sensor-based cross attention module to utilize the LiDAR point as query, and its neighbored projected pixel coordinates and RGB values as keys and values for the fusion feature computation. Piergiovanni et al. [35] studied a general 4D detection framework for both RGB images and LiDAR point clouds in a time series. To deal with the sparsity of the LiDAR point cloud, Yin et al. [36] attempted to generate 3D virtual points of a targeted object with the guidance of the instance segmentation result predicted from the RGB image.

### 2.2. Semi-Supervised 3D Object Detection

Semi-supervised learning (SSL) is a classical problem in machine learning and deep learning [37]. Compared with the booming development of FSOD-3D and the rapid development of SSOD-2D, relatively fewer works on SSOD-3D have been published in academia. However, it is a challenging and meaningful problem for both industry and academia. First, unlike FSOD-3D, SSOD-3D needs to both consider how to generate reliable pseudo or weak labels from unlabeled point clouds, and how to exploit pseudo labels with uncertain quality for 3D detector training. Second, a classical SSL framework is difficult to directly use in SSOD-3D. For the labeled data $\mathbf{x}_i$ and unlabeled data $\mathbf{x}_j$, traditional SSL theory [37] emphasises their similarity $w(\mathbf{x}_i, \mathbf{x}_j) \in [0, 1]$, and constructs a manifold regularization term for SSL optimization. However, in SSOD-3D, point clouds $\mathbf{x}_i$ and $\mathbf{x}_j$ are collected from different places and have different data distributions; thus, it is difficult to measure their similarity. Third, as the sparse and unstructured LiDAR point cloud contains fewer texture features than the dense and structured RGB image, it is more difficult for SSOD-3D to extract salient prior knowledge of the targeted object than it is for SSOD-2D.

Some insightful works of SSOD-3D are discussed hereafter. Tang and Lee [9] exploited the weak label (i.e., the 2D bounding box in RGB image) of an unlabeled 3D point cloud to train a 3D detector. The weak label is generated via the 2D object detector. To compute 3D detection loss with the weak label, the predicted 3D bounding box is projected onto the image plane. After that, 3D detection loss is converted into 2D detection loss. This

method requires both RGB imaging and point cloud, and it works for both RGB-D camera and LiDAR-camera systems. Xu et al. [38] adaptively filtered the incorrect 3D object in the unlabeled data with a statistical and adaptive confidence threshold, and added the remaining predicted 3D objects into the 3D object database for 3D object detector training in the next iteration.

Mean teacher [39] is a common SSL paradigm in SSOD-3D. It consists of teacher and student detectors. For one unlabeled data, its pseudo label is generated from the teacher detector and used to supervise the student detector. Zhao et al. [10] were the first to utilize the mean teacher framework [39] for SSOD-3D. For the unlabeled data $\mathbf{x}$, they generated its pseudo label from the teacher detector and constructed consistency loss to minimize the difference between the pseudo label and result predicted by the student detector with data augmentation on $\mathbf{x}$. After that, parameters in the teacher detector were updated with the trained student detector via exponential moving average (EMA). Some current literature [11,40,41] has attempted to improve the previous work [10]. Wang et al. [11] focused on how to remove incorrect annotations from the predicted label $\mathbf{l}$ with multi-thresholds of object confidence, class, and 3D IoU. Wang et al. [40] attempted to generate accurate predicted labels with temporal smoothing. The teacher 3D detector predicted multi-frame labels from the multi-frame LiDAR data. After that, temporal GNN was used to generate the accurate labels at the current frame from these multi-frame labels. Park et al. [41] exploited a multi-task (i.e., 3D and 2D object detection) teacher detector to establish multi-task guided consistency loss for supervision. It works on a LiDAR-camera system. Sautier et al. [42] presented a self-supervised distillation method to pre-train the backbone network in a 3D object detector, with the guidance of super-pixel segmentation results, on an RGB image.

Some researchers consider that weak label is convenient and time-efficient for annotation, and study weak-supervised 3D object detection (WSOD-3D). Meng et al. [43] proposed a weak and fast 3D annotation procedure to generate a 3D cylindrical bounding box by clicking the object center in an RGB image. With the cylindrical label, they converted SSOD-3D as WSOD-3D and provided a two-stage training scheme for the 3D object detector. Qin et al. [44] designed an unsupervised 3D proposal generation method to obtain the 3D bounding box with anchor size, using the normalized point cloud density. Peng et al. [45] presented a WSOD-3D method for a monocular camera, which utilizes the alignment constraint of predicted 3D proposal and LiDAR points for the weak supervision. Xu et al. [46] dealt with WSOD-3D under the condition that the position-level annotations are known. A virtual scene with GT annotation is constructed with the known object centers. Then, samples in the real scene with weak labels and in the virtual scene with GT labels are both used for detector training.

### 2.3. Discussions

From the above analysis, most of the current study of SSOD-3D emphasises improving the quality of pseudo labels. Two common schemes are utilized: (i) label filter [10,11,38,41] and (ii) temporal smoothing [40]. However, both of these have space for improvement. The label filter scheme is not time-efficient enough to search for the optimal filter thresholds. The temporal smoothing scheme needs multi-frame LiDAR point clouds with the accurate sensor pose information; this need is difficult to satisfy in some actual situations. Although the weak label (i.e., 2D bounding box [9,45], 3D cylindrical bounding box [43] and 3D center position [46]) is easier for annotation than the standard label (i.e., 3D bounding box), it still costs time and human resources for the amount of unlabeled data that needs annotation in the context of autonomous driving. Therefore, an effective SSOD-3D method is still required.

## 3. Proposed Semi-Supervised 3D Object Detection

### 3.1. Problem Statement

SSOD-3D is a training framework to learn baseline detection from the labeled and unlabeled datasets $\mathbb{X}_l$ and $\mathbb{X}_u$. The baseline detector is the arbitrary 3D object detector based on the LiDAR point cloud. Let $\mathbf{w}$ be the parameter set of baseline detector. SSOD-3D aims to learn $\mathbf{w}^*$ with higher generalization ability.

Some symbols are discussed here. Let $\mathbf{x}_i = \{\mathbf{P}_i\}$ be the $i$-th training sample where $\mathbf{x}_i \in \mathbb{X}_l$ or $\mathbf{x}_i \in \mathbb{X}_u$ means that it is ground truth (GT), labeled or not. $\mathbf{P}_i$ is $[N_i, 4]$ LiDAR point cloud where $N_i$ is number of LiDAR points. It contains the 3D position and reflected intensity of the LiDAR point cloud. Let $\mathbf{l}_i = \{l_{ij}\}_{j=1}^{n_i}$ be the 3D object label of $\mathbf{x}_i$. $n_i$ is the object number. $l_{ij}$ represents the 3D bounding box of the $j$-th object using the parameter vector of the 3D bounding box [28]. $\mathbf{l}_i$ is the pseudo or GT label if $\mathbf{x}_i \in \mathbb{X}_u$ or $\mathbf{x}_i \in \mathbb{X}_l$. Let $\mathbf{l}_i^p = f(\mathbf{x}_i; \mathbf{w})$ be the output of the 3D object detector with the input of $\mathbf{x}_i$ and weight of $\mathbf{w}$. $\mathbf{l}_i^p = \{l_{ik}^p\}_{k=1}^{n_i}$ is the pseudo label of $\mathbf{x}_i$.

### 3.2. Previous Semi-Supervised 3D Object Detection

Before illustrating the proposed SPSL-3D, we briefly revisit the previous SSOD-3D approach [10]. The pipeline of the previous SSOD-3D is presented in Figure 1a. For the 3D object detector with high generalization ability, its prediction results from the unlabeled sample $\mathbf{x}_i$ and its augmented sample $\mathbf{A}(\mathbf{x}_i)$ are both consistent and closed to the GT labels. Based on this analysis, as the unlabeled sample does not have annotation, $L_{cons}$ was proposed to minimize the difference in labels predicted from $\mathbf{x}_i$ and $\mathbf{A}(\mathbf{x}_i)$. $\mathbf{A}(\mathbf{x}_i)$ is the affine transformation on $\mathbf{P}_i$ of $\mathbf{x}_i$, which contains scaling, X/Y-axis flipping, and Z-axis rotating operations. $L_{cons}$ is the core in this scheme [10], for this loss can update the weights in the 3D object detector via back-propagation. The current SSOD-3D [10,11,38,41,47] optimizes $\mathbf{w}^*$ by minimizing the function as:

$$\mathcal{L}_{\text{SSOD-3D}}(\mathbf{w}) = \sum_{i=1}^{N_l} \|L_{3d}(\mathbf{l}_i^p, \mathbf{l}_i)\|_1 + L_{cons}, \mathbf{x}_i \in \mathbb{X}_l \tag{1}$$

$$L_{cons}(\mathbf{w}) = \sum_{j=1}^{N_u} \|L_{3d}(\mathbf{l}_j^p, \mathbf{A}^{-1}(\mathbf{l}_{j,\text{Aug}}^p))\|_1, \mathbf{x}_j \in \mathbb{X}_u \tag{2}$$

$$\mathbf{l}_j^p = f(\mathbf{x}_j; \mathbf{w}), \ \mathbf{l}_{j,\text{Aug}}^p = f(\mathbf{A}(\mathbf{x}_j); \mathbf{w})$$

where $L_{3d}(\mathbf{l}_i^p, \mathbf{l}_i)$ is the common 3D object detection loss of each detected object [20,28]. It is represented as $[n_i, 1]$ vector to describe the detection loss of each object. With the inverse affine transformation $\mathbf{A}^{-1}(\cdot)$, $\mathbf{A}^{-1}(\mathbf{l}_{i,\text{Aug}}^p)$ is obtained with the same reference coordinate system as in $\mathbf{l}_i^p$. In the end, we also provide discussion of the relation of a previous SSOD-3D, and traditional SSL theory is further discussed in Appendix A.1.

### 3.3. Self-Paced Semi-Supervised Learning-Based 3D Object Detection

The main challenge of consistency loss in Equation (2) is that the quality of the pseudo label $\mathbf{l}_j^p$ is uncertain. As $N_u > N_l$, if $\mathbf{l}_j^p$ is noisy or even incorrect, the baseline detector with the optimized parameter set $\mathbf{w}^*$ tends to detect 3D objects with low localization accuracy. To deal with this problem, we needs to evaluate the reliability weight $\mathbf{v}_j$ of $\mathbf{l}_j^p$, where $\mathbf{v}_j = (v_{j1}, \cdots, v_{jk}, \cdots, v_{jn_j})^T$ is a vector to reflect the reliability score of objects in $\mathbf{l}_j^p$ ($v_{jk} \in [0, 1]$). In the training stage, unreliable pseudo labels are filtered out with $\mathbf{v}_j$. However, determining $\mathbf{v}_j$ is a crucial problem.

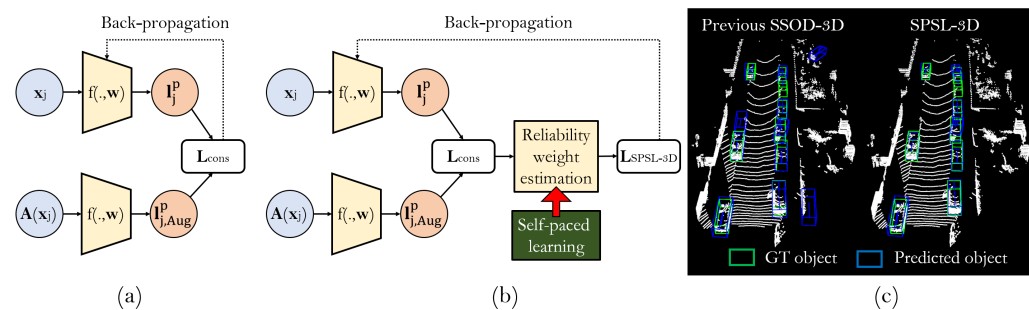

**Figure 1.** Difference between the previous SSOD-3D [10] and the proposed SPSL-3D. (**a**) Consistency loss [10]. (**b**) Loss in SPSL-3D. This emphasizes the quality of pseudo label $\mathbf{l}_j^p$, adjusting the reliability weight of object in $\mathbf{l}_j^p$ and thus enhancing the generalization ability of the baseline detector. (**c**) Improvement of SPSL-3D in 3D object detection.

One naive idea is to adjust the reliability weight $\mathbf{v}_j$ with the guidance of the consistency loss of $\mathbf{l}_j^p$. If the consistency loss of $\mathbf{l}_j^p$ enlarges, the pseudo labels $\mathbf{l}_j^p$ are unreliable. Based on this idea, we exploit the theory of SPL [18] to construct the mathematical relation of $\mathbf{v}_j$ to $L_{3d}(\mathbf{l}_j^p, \mathbf{A}^{-1}(\mathbf{l}_{j,\text{Aug}}^p))$, and propose a novel SSOD-3D framework, SPSL-3D, in this paper. Its pipeline is presented in Figure 1b. SPSL-3D optimizes $\mathbf{w}^*$ by minimizing the function as:

$$\mathcal{L}_{\text{raw}}(\mathbf{w},\mathbf{v},\lambda_l,\lambda_u) = \sum_{i=1}^{N_l} \mathbf{v}_i^T L_{3d}(\mathbf{l}_i^p, \mathbf{l}_i) + \sum_{j=1}^{N_u}\left(\mathbf{v}_j^T L_{3d}(\mathbf{l}_j^p, \mathbf{A}^{-1}(\mathbf{l}_{j,\text{Aug}}^p)) + f_u(\{\mathbf{v}_j\}_{j=1}^{N_u}, \lambda_u)\right),$$
$$\mathbf{v} = \{\mathbf{v}_j\}_{j=1}^{N_u}, \mathbf{x}_i \in \mathbb{X}_i, \mathbf{x}_j \in \mathbb{X}_j \tag{3}$$

$$\lambda_u = \frac{e}{E}\max(L_{3d}(\mathbf{l}_j^p, \mathbf{A}^{-1}(\mathbf{l}_{j,\text{Aug}}^p))) + \left(1 - \frac{e}{E}\right)\text{mean}(L_{3d}(\mathbf{l}_j^p, \mathbf{A}^{-1}(\mathbf{l}_{j,\text{Aug}}^p))) \tag{4}$$

where $\lambda_u$ is age parameter to control the learning pace [48]. Let the current epoch and maximum training epoch be $e$ and $E$. Furthermore, $f_u(\{\mathbf{v}_j\}_{j=1}^{N_u}, \lambda_u)$ is a self-paced regularization term [48] for the unlabeled sample:

$$f_u(\{\mathbf{v}_j\}_{i=1}^{N_u}, \lambda_u) = -\lambda_u \sum_{j=1}^{N_u}\sum_{k=1}^{n_j}\left(-\frac{1}{2}v_{jk}^2 + v_{jk}\right) \tag{5}$$

However, in deep learning, $\mathbf{w}$ contains lots of parameters, so it is difficult to directly optimize Equation (3). As the modern deep neural network (DNN) is trained with a batch of data the size of $B_l + B_u$ [49], the loss of SPSL-3D is simplified as:

$$\mathcal{L}_{\text{SPSL-3D}}(\mathbf{w},\mathbf{v},\lambda_l,\lambda_u) = \sum_{i=1}^{B_l}\left(\mathbf{v}_i^T L_{3d}(\mathbf{l}_i^p, \mathbf{l}_i)\right)$$
$$+ \sum_{j=1}^{B_u}\left(\mathbf{v}_j^T L_{3d}(\mathbf{l}_j^p, \mathbf{A}^{-1}(\mathbf{l}_{j,\text{Aug}}^p)) - \lambda_u\sum_{k=1}^{n_j}\left(-\frac{1}{2}v_{jk}^2 + v_{jk}\right)\right), \tag{6}$$
$$\mathbf{v} = \{\mathbf{v}_j\}_{j=1}^{B_u}, \mathbf{x}_i \in \mathbb{X}_i, \mathbf{x}_j \in \mathbb{X}_j$$

An alternative optimization scheme is used to optimize $\mathbf{w}$ and $\mathbf{v}$. With the fixed $\mathbf{w}$, $\mathbf{v}$ needs to be optimized. The closed-form solution is obtained as Equation (7) via $\partial\mathcal{L}_{\text{SPSL-3D}}/\partial\mathbf{v} = 0$. For a vector $L$, $[L]_k$ is its $k$-th element. With the fixed $\mathbf{v}$, $\mathbf{w}$ is optimized in Equation (6) with the deep learning-based optimizer (i.e., Adam and SGD).

$$v_{jk} = \begin{cases} 1 - \dfrac{[L_{3d}(\mathbf{l}_j^p, \mathbf{A}^{-1}(\mathbf{l}_{j,\text{Aug}}^p))]_k}{\lambda_u}, & [L_{3d}(\mathbf{l}_j^p, \mathbf{A}^{-1}(\mathbf{l}_{j,\text{Aug}}^p))]_k < \lambda_u \\ 0, & [L_{3d}(\mathbf{l}_j^p, \mathbf{A}^{-1}(\mathbf{l}_{j,\text{Aug}}^p))]_k \geq \lambda_u \end{cases} \tag{7}$$

Intuitive explanation of Equation (7) is discussed here. For the $k$-th object in the sample with the pseudo label, if its loss is larger than $\lambda_u$, it is regarded as an unreliable label and cannot be used. If its loss is smaller than $\lambda_u$, SPSL-3D evaluates its reliability score with its consistency loss. SPSL-3D emphasizes the most reliable pseudo label in the training stage to enhance the robustness of the baseline detector. When epoch $e$ grows, $\lambda_u$ increases (seen Equation (4)), meaning that SPSL-3D enlarges the size of the unlabeled samples for training, thus improving the generalization ability of baseline detector. The improvement can be found in Figure 1c.

### 3.4. Improving SPSL-3D with Prior Knowledge

From Equation (7), SPSL-3D can adaptively adjust the reliability weight of pseudo label using its 3D object detection loss. In fact, the reliability weight of pseudo label is not only dependent on consistency loss, but also dependent on the prior knowledge in the LiDAR point cloud and RGB image provided by the LiDAR-camera system. If the LiDAR point cloud or image feature of one predicted object is not salient, its pseudo label is not reliable. Based on this analysis, to further enhance the performance of PSPL-3D with information from the LiDAR point cloud and RGB image, we propose a prior knowledge-based SPSL-3D named PSPSL-3D, which is presented in Figure 2.

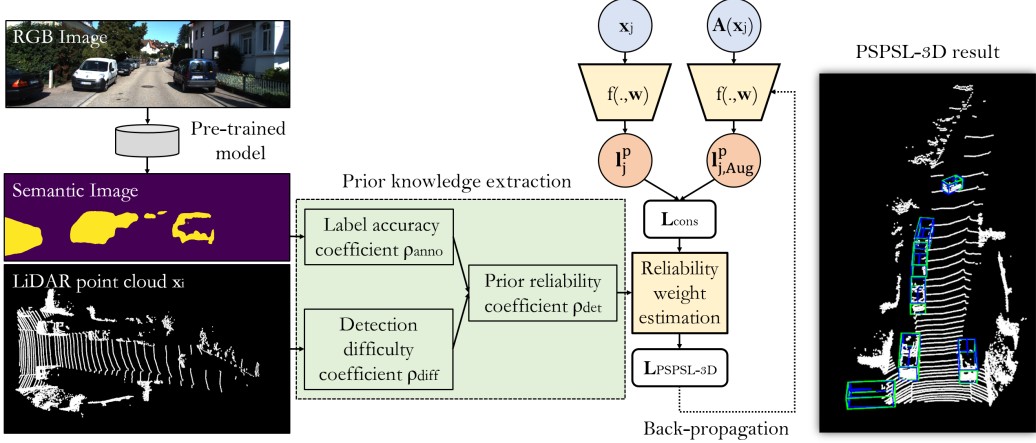

**Figure 2.** Framework of the proposed PSPSL-3D. It can evaluate the reliability weight of a pseudo label from prior knowledge extracted from the LiDAR point cloud and RGB image.

In PSPSL-3D, we attempt to represent the reliability of the pseudo label with the LiDAR point cloud and RGB image. For the $k$-th object $o_{jk}$ in $\mathbf{l}_j^p$, its prior reliability coefficient is modeled as $\rho_{\text{det},jk} = \rho_{\text{diff},jk} \cdot \rho_{\text{anno},jk}$. It consists of the detection difficulty coefficient $\rho_{\text{diff},jk} \in [0,1]$ and the label accuracy coefficient $\rho_{\text{anno},jk} \in [0,1]$. The motivation of designing $\rho_{\text{det},jk}$ is to constrain $v_{jk}$ with both 3D detection loss and prior knowledge from the RGB image and LiDAR point cloud.

Due to the LiDAR mechanism, $\rho_{\text{diff},jk}$ is mainly dependent on the occlusion and resolution of $o_{jk}$ in the LiDAR point cloud. However, due to the complex situation of 3D object in the real traffic situation, it is difficult to model the relationship between the occlusion, resolution, and detection difficulty of $o_{jk}$. An approximate solution is provided here. Following the thought in the literature [50], a statistical variable $m_{jk}$ is used as the LiDAR point number inside the 3D bounding box of $o_{jk}$ to describe $\rho_{\text{diff},jk}$:

$$\rho_{\text{diff},jk} = \frac{\min(m_{jk}, m_{th}(c_{jk}))}{m_{th}(c_{jk})} \tag{8}$$

where $c_{jk}$ is the category (i.e., car, pedestrian, cyclist) of $o_{jk}$, and $m_{th}(c_{jk})$ is the minimal threshold of the LiDAR point number of the corresponding category. For the actual implementation, $m_{th}(c_{ik})$ is a statistic variable from $\mathbb{X}_l$, discussed in Section 4.2. If $o_{jk}$ has higher resolution and less occlusion, $m_{jk}$ is closer and even higher than $m_{th}(c_{jk})$, so that $\rho_{\text{diff},jk}$ is closer to 1. The 3D detection difficulty of $o_{jk}$ is largely decreased.

Then, $\rho_{\text{anno},jk}$ is discussed. As GT is unknown, we attempt to evaluate the annotation accuracy indirectly. On the one hand, as for the current 3D detection method, a confidence score $s_{jk}^p \in [0, 1]$ of $o_{jk}$ is supervised with the 3D IoU of the predicted and GT 3D bounding box [21,26]. $s_{jk}^p$ can be used to describe $\rho_{\text{anno},jk}$. On the other hand, a semantic segmentation map predicted from RGB image also contains annotation information. As the RGB image is more dense than the LiDAR point cloud, semantic segmentation on the RGB image is more accurate than the semantic segmentation on the LiDAR point cloud. Projecting the 3D bounding box of $o_{jk}$ onto the image plane generates a 2D bounding box $B_{jk}$. Its pixel area is $A_{jk}$. The pixel area of the semantic map of $c_{jk}$ inside $B_{jk}$ is $S_{jk}$. If the predicted 3D bounding box of $o_{jk}$ is accurate, $s_{jk}^p$ and ratio of $S_{jk}$ and $A_{jk}$ are closer to 1. Due to the occlusion of object, $S_{jk}$ is not accurate enough. Thus, the arithmetic mean of $s_{jk}^p$ and the pixel area ratio is used to describe $\rho_{\text{anno},jk}$:

$$\rho_{\text{anno},jk} = \sqrt{s_{ik}^p \cdot \frac{S_{ik}}{A_{ik}}} \tag{9}$$

From the above discussion, prior knowledge is not directly extracted from the RGB image, for the RGB feature of the targeted object is affected by shadow and blur in the complex traffic scenario. Compared with the RGB image, the semantic segmentation map is more stable to reflect the location information of a targeted object. Thus, the semantic feature is used to describe $\rho_{\text{anno},jk}$.

After obtaining $\rho_{\text{det},jk}$, a scheme designed to constrain $v_{jk}$ with $\rho_{\text{det},jk}$ is required. Referring to the thought in self-paced curriculum learning [51], the interval of $v_{jk}$ can be constrained from $[0, 1]$ to $[0, \rho_{\text{det},jk}]$. This means that the interval of $v_{jk}$ is dependent on its prior detection coefficient $\rho_{\text{det},jk}$. To achieve this scheme, $v_{jk}$ in Equation (6) is replaced with $v_{jk}/\rho_{\text{det},jk}$, and the loss function of PSPSL-3D is presented as:

$$\begin{aligned}
\mathcal{L}_{\text{PSPSL-3D}}(\mathbf{w}, \mathbf{v}, \lambda_l, \lambda_u) &= \sum_{i=1}^{B_l} \left( \mathbf{v}_i^T L_{3d}(\mathbf{l}_i^p, \mathbf{l}_i) \right) \\
&+ \sum_{j=1}^{B_u} \left( \mathbf{v}_j^T L_{3d}(\mathbf{l}_j^p, \mathbf{A}^{-1}(\mathbf{l}_{j,\text{Aug}}^p)) - \lambda_u \sum_{k=1}^{n_j} \left( -\frac{1}{2} \left( \frac{v_{jk}}{\rho_{\text{det},jk}} \right)^2 + \frac{v_{jk}}{\rho_{\text{det},jk}} \right) \right), \\
\mathbf{v} &= \{\mathbf{v}_j\}_{j=1}^{B_u}, \mathbf{x}_i \in \mathbb{X}_i, \mathbf{x}_j \in \mathbb{X}_j
\end{aligned} \tag{10}$$

As the same in Section 3.2, the alternative optimization scheme is used to find the optimal $\mathbf{w}$ and $\mathbf{v}$. The close-formed solution of $\mathbf{v}$ is shown in Equation (11) via $\partial \mathcal{L}_{\text{PSPSL-3D}}/\partial \mathbf{v} = 0$. The procedure of PSPSL-3D is summarized in Algorithm 1.

$$v_{jk} = \begin{cases}
\rho_{\text{det},jk} \cdot \left( 1 - \dfrac{[L_{3d}(\mathbf{l}_j^p, \mathbf{A}^{-1}(\mathbf{l}_{j,\text{Aug}}^p))]_k}{\lambda_u} \right), & [L_{3d}(\mathbf{l}_j^p, \mathbf{A}^{-1}(\mathbf{l}_{j,\text{Aug}}^p))]_k < \lambda_u \\
0, & [L_{3d}(\mathbf{l}_j^p, \mathbf{A}^{-1}(\mathbf{l}_{j,\text{Aug}}^p))]_k \geq \lambda_u
\end{cases} \tag{11}$$

---

**Algorithm 1** Proposed SPSL-3D and PSPSL-3D framework for SSOD-3D.

---

**Inputs**: Baseline detector, maximum epoch $E$, datasets $\mathbb{X}_l$ and $\mathbb{X}_u$, batch sizes $B_l$ and $B_u$;
**Parameters**: Baseline detector weight $\mathbf{w}$, current epoch $e$, sample weight $\mathbf{v}$, age parameters $\lambda_l$ and $\lambda_u$;
**Output**: Optimal 3D detector weight $\mathbf{w}^*$

  1: Pre-training baseline 3D detector in $\mathbb{X}_l$ obtains $\mathbf{w}_0$
  2: Let $e = 1$ and $\mathbf{w} = \mathbf{w}_0$
  3: **while** $e \leq E$ **do**
  4:     Let $k = 1$
  5:     **while** $k \leq \mathrm{N}_l$ **do**
  6:         $\{\mathbf{x}_i, \mathbf{l}_i\}_{i=1}^{B_l} = \mathrm{DataLoader}(\mathbb{X}_l, B_l)$, $\{\mathbf{x}_j\}_{j=1}^{B_u} = \mathrm{DataLoader}(\mathbb{X}_u, B_u)$
  7:         Computing $\lambda_u$ with $e$ and $E$ via Equation (4)
  8:         **if** SPSL-3D is exploited **then**
  9:             Optimizing $\mathbf{v}$ using $\mathbf{w}$, $\lambda_u$ via Equation (7)
10:             Optimizing $\mathbf{w}$ with $\mathbf{v}$ via Equations (6)
11:         **end if**
12:         **if** PSPSL-3D is exploited **then**
13:             Computing $\rho_{\mathrm{det},jk}$ via Equations (8) and (9)
14:             Optimizing $\mathbf{v}$ using $\mathbf{w}$, $\rho_{\mathrm{det},ik}$, $\lambda_u$ via Equation (11)
15:             Optimizing $\mathbf{w}$ with $\mathbf{v}$ via Equation (10)
16:         **end if**
17:         $k = k + B_l$
18:     **end while**
19:     $e = e + 1$
20: **end while**
21: Return $\mathbf{w}^*$ as $\mathbf{w}$.

---

## 4. Experiments

### 4.1. Dataset and Configuration

The classical outdoor KITTI dataset [19] is exploited to evaluate 3D detection performance in the outdoor traffic scenario. It contains (i) training dataset $\mathbb{D}_{\mathrm{train}}$ with 3712 samples, (ii) validation dataset $\mathbb{D}_{\mathrm{val}}$ with 3769 samples, and (iii) testing dataset $\mathbb{D}_{\mathrm{test}}$ with 7518 samples. All of them have GT annotation. The LiDAR point cloud and RGB image are provided in each sample. As the raw KITTI dataset does not contain semantic segmentation images, we generate semantic maps with four categories (i.e., car, pedestrian, bicycle, background) using the pre-trained deeplab v3 [52]. Three categories (i.e., car, pedestrian, cyclist) of targeted object are considered in the following experiments. To verify the performance of SSOD-3D methods, a semi-supervised condition is established in the experiments. $\mathbb{D}_{\mathrm{train}}$ is divided as $\mathbb{X}_l$ and $\mathbb{X}_u$ where $\mathbb{X}_l \cup \mathbb{X}_u = \mathbb{D}_{\mathrm{train}}$ and $\mathbb{X}_l \cap \mathbb{X}_u = \varnothing$. GT labels in $\mathbb{X}_u$ are disabled in the training stage. $\mathbb{X}_u$ is regarded as unlabeled dataset. Let $R_{\mathrm{anno}} = \mathrm{N}_l / (\mathrm{N}_l + \mathrm{N}_u)$ be the labeled ratio. In the following experiments, to evaluate the performance of SSOD-3D comprehensively, we set various training situations with the different labeled ratios, from 4% (hard SSL case ) to 64% (easy SSL case). Specifically, $R_{\mathrm{anno}}$ is set as 4%, 8%, 16%, 32%, 64%, respectively. Furthermore, we mainly focus on the comparison results in the hard SSL case ($R_{\mathrm{anno}} \leq 16\%$).

To measure the results of SSOD-3D methods, SSOD-3D methods are trained on $\mathbb{D}_{\mathrm{train}}$, and then evaluated on $\mathbb{D}_{\mathrm{val}}$. 3D average precision (AP) is the main metric for comparison. In order to further evaluate the different SSOD-3D methods, a bird's eye view (BEV) AP and 3D recall rate are also provided. The IoU thresholds of 3D and BEV objects are 0.7 (car) and 0.5 (pedestrian and cyclist). As object label in KITTI dataset has three levels (i.e., easy, moderate, hard), these metrics of all level objects are provided for the comprehensive comparison.

The proposed SPSL-3D, PSPSL-3D needs a baseline detector. Voxel RCNN [21] is selected as the baseline 3D detector as it has simple detector architecture and fast and accurate inference performance. THe optimizer, learning rate policy, and hyper-parameters

in $\mathbf{A}(\cdot)$ are default [21]. As the proposed method is implemented on a single Nvidia GTX 3070, the batch size is set as 2, where $B_l = 1$ and $B_u = 1$. $\mathbf{w}_0$ is needed for SPSL-3D and PSPSL-3D. This is obtained by per-training Voxel R-CNN on $\mathbb{X}_l$ with 80 epochs. The maximum epoch $E$ is related to $R_{\text{anno}}$. Fine-tuning experience shows that better results are achieved when $E = 30$ if $R_{\text{anno}} \leq 16\%$ and $E = 50$ if $R_{\text{anno}} > 16\%$. In the actual training stage, data augmentation contains not only 3D affine transformation on the LiDAR point cloud, but also the cut-and-paste operation [4]. This operation aims to increase the object number in $\mathbf{x}_i$ by putting extra 3D objects with GT annotation into the point cloud $\mathbf{P}_i$. 3D objects with GT labels are stored in the object bank before training. To prevent data leakage, the object bank should be built only on $\mathbb{X}_l$ instead of $\mathbb{D}_{\text{train}}$.

Current SSOD-3D methods are selected for comparison. SESS [10] is the first to utilize the mean-teacher SSL framework [39] in SSOD-3D. SESS is enhanced with the multi-threshold label filter (LF) proposed in work [11], and the improved method is named as 3DIoUMatch. UDA [53] is a classical SSL framework. It exploits the consistency loss $L_{\text{cons}}$ for supervision. We consider that it can work for SSOD-3D. As curriculum learning (CL) [54] is a fundamental part of SPL theory [18], CL can also be used in SSOD-3D, so that method UDA+CL is designed. As unlabeled data increase learning uncertainty, UDA+CL tries to learn $\mathbf{w}$ with increasing unlabeled data. For $L_{\text{cons}}$ in Equation (3), $N_l$ is replaced with $\lfloor \alpha(e) N_l \rfloor$ where $\alpha(e) = e/E$. These methods also require a baseline 3D detector. For a fair comparison, Voxel RCNN [21] is marked as Baseline and used as the baseline 3D detector for all SSOD-3D methods. The above mentioned methods are trained in the same condition. As for the SSOD-3D method on the LiDAR point cloud, only the open-source code of 3DIoUMatch [11] is provided (https://github.com/THU1 7cyz/3DIoUMatch-PVRCNN, accessed on 1 July 2022). Other methods are implemented by authors on the open-source FSOD-3D framework OpenPCDet (https://github.com/open-mmlab/OpenPCDet, accessed on 1 March 2022).

*4.2. Comparison with Semi-Supervised Methods*

This experiment investigates the comparison results of the proposed PSPSL-3D methods with the current SSOD-3D methods. The results of all SSOD-3D methods at $R_{\text{anno}} = 4\%, 8\%, 16\%, 32\%, 64\%$ are presented in Tables 1–5. In these tables, gain from baseline means the improvement of the proposed PSPSL-3D over the baseline method. For the baseline, its 3D mAPs of all categories are dramatically increased from $R_{\text{anno}} = 4\%$ to 8%, suggesting the large potential improvement of SSOD-3D. The 3D mAPs of baseline increased slowly when $R_{\text{anno}} \geq 16\%$, and the improvements of SSOD-3D methods are also relatively small. The 3D mAP of 3DIoUMatch [11] is higher than SESS [10], as the multi-threshold-based label filter in 3DIoUMatch can remove some incorrectly predicted labels. The 3D mAP of UDA+CL [54] is higher than UDA [53], as the curriculum can reduce certain instances of overfitting of easily detected objects in the training stage. In most cases, the proposed SPSL-3D is superior to SESS [10] and UDA [53], because it exploits SPL theory [18] to filter incorrect and too difficult labeled and unlabeled training samples adaptively. It is noticed that the proposed PSPSL-3D has higher 3D mAPs than other current methods because it adds prior knowledge from the RGB image and LiDAR point cloud as self-paced regularization terms in SPSL-3D to achieve robust and accurate learning results. It is found that 3DIouMatch [11] has better performance than other previous methods. Compared with 3DIoUMatch, the proposed PSPSL-3D makes a significant improvement in 3D cyclist detection and also has a certain improvement in 3D pedestrian detection, because the prior knowledge from the LiDAR point cloud and RGB image is beneficial to modeling the reliability of objects with a relatively small size. Therefore, it is concluded that the proposed SPSL-3D and PSPSL-3D benefit SSOD-3D in a LiDAR-camera system.

**Table 1.** 3D AP of current SSOD-3D methods in KITTI validation dataset at $R_{anno} = 4\%$.

| 4% Labeled | 3D AP of Car | | | 3D AP of Pedestrian | | | 3D AP of Cyclist | | |
|---|---|---|---|---|---|---|---|---|---|
| Methods | Easy | Moderate | Hard | Easy | Moderate | Hard | Easy | Moderate | Hard |
| Baseline [21] | 36.88% | 32.05% | 30.03% | 19.71% | 16.76% | 16.40% | 13.84% | 13.45% | 13.50% |
| SESS [10] | 39.07% | 33.81% | 32.20% | 15.02% | 12.35% | 12.37% | 13.98% | 12.80% | 12.75% |
| UDA [53] | 38.43% | 35.94% | 31.57% | 20.25% | 19.81% | 17.06% | 15.42% | 7.67% | 7.79% |
| 3DIoUMatch [11] | 47.73% | 41.15% | 39.33% | 21.53% | 22.01% | 18.04% | 18.33% | 14.57% | 13.32% |
| UDA+CL [54] | 40.21% | 36.17% | 32.19% | 22.52% | 19.34% | 18.11% | 15.97% | 10.12% | 9.09% |
| SPSL-3D | 46.98% | 40.87% | 38.16% | **25.58%** | **21.71%** | 18.05% | 19.27% | 13.70% | 13.73% |
| PSPSL-3D | **52.24%** | **42.52%** | **40.30%** | 24.83% | 21.22% | **21.17%** | **20.57%** | **15.15%** | **13.79%** |
| Gain from baseline | +15.36% | +10.47% | +10.27% | +5.12% | +4.46% | +4.77% | +6.73% | +1.70% | +0.29% |

**Table 2.** 3D AP of current SSOD-3D methods in KITTI validation dataset at $R_{anno} = 8\%$.

| 8% Labeled | 3D AP of Car | | | 3D AP of Pedestrian | | | 3D AP of Cyclist | | |
|---|---|---|---|---|---|---|---|---|---|
| Methods | Easy | Moderate | Hard | Easy | Moderate | Hard | Easy | Moderate | Hard |
| Baseline [21] | 74.64% | 63.57% | 57.25% | 37.79% | 31.96% | 30.53% | 50.73% | 34.51% | 30.18% |
| SESS [10] | **77.70%** | 64.14% | 59.66% | 41.11% | 33.79% | 33.17% | 50.92% | 33.24% | 32.67% |
| UDA [53] | 76.02% | 64.88% | 61.47% | 37.90% | 32.91% | 31.92% | 49.87% | 35.20% | 30.07% |
| 3DIoUMatch [11] | 76.12% | 65.16% | 58.19% | **43.28%** | 36.56% | 32.17% | 48.82% | 34.85% | **34.12%** |
| UDA+CL [54] | 76.14% | 65.15% | 62.59% | 38.08% | 33.47% | 31.99% | 50.35% | 35.19% | 30.25% |
| SPSL-3D | 76.40% | 65.79% | **63.70%** | 39.54% | 35.01% | 30.78% | 52.15% | 32.23% | 31.18% |
| PSPSL-3D | 77.03% | **66.03%** | 63.28% | 42.24% | **37.14%** | **32.14%** | **55.14%** | **38.76%** | 32.69% |
| Gain from baseline | +2.39% | +2.46% | +6.03% | +4.45% | +5.18% | +1.61% | +4.41% | +4.25% | +2.51% |

**Table 3.** 3D AP of current SSOD-3D methods in KITTI validation dataset at $R_{anno} = 16\%$.

| 16% Labeled | 3D AP of Car | | | 3D AP of Pedestrian | | | 3D AP of Cyclist | | |
|---|---|---|---|---|---|---|---|---|---|
| Methods | Easy | Moderate | Hard | Easy | Moderate | Hard | Easy | Moderate | Hard |
| Baseline [21] | 86.52% | 75.06% | 68.98% | 50.17% | 45.96% | 43.09% | 64.72% | 47.99% | 47.05% |
| SESS [10] | 85.74% | 75.25% | 68.52% | 51.16% | 47.36% | 45.36% | 63.22% | 48.03% | 46.54% |
| UDA [53] | 86.21% | 75.73% | 73.90% | 50.42% | 48.33% | 43.34% | 66.12% | 47.39% | 46.88% |
| 3DIoUMatch [11] | **87.41%** | 76.41% | 74.57% | 52.39% | 49.33% | **45.59%** | 65.30% | 48.88% | 47.35% |
| UDA+CL [54] | 86.67% | 75.97% | 74.55% | 50.22% | 48.36% | 43.58% | 66.57% | 48.85% | 47.74% |
| SPSL-3D | 86.98% | 76.14% | 74.65% | 50.67% | 48.76% | 43.72% | 73.43% | 48.77% | 48.20% |
| PSPSL-3D | 87.18% | **76.49%** | **74.66%** | **52.41%** | **49.64%** | 44.29% | **74.71%** | **49.68%** | **49.28%** |
| Gain from baseline | +0.66% | +1.43% | +5.68% | +2.24% | +3.68% | +1.20% | +9.99% | +1.69% | +2.23% |

**Table 4.** 3D AP of current SSOD-3D methods in KITTI validation dataset at $R_{anno} = 32\%$.

| 32% Labeled | 3D AP of Car | | | 3D AP of Pedestrian | | | 3D AP of Cyclist | | |
|---|---|---|---|---|---|---|---|---|---|
| Methods | Easy | Moderate | Hard | Easy | Moderate | Hard | Easy | Moderate | Hard |
| Baseline [21] | 87.20% | 77.14% | 75.44% | 59.13% | 52.55% | 50.49% | 73.85% | 54.21% | 52.26% |
| SESS [10] | 87.67% | 76.91% | 75.25% | 62.61% | 55.74% | 49.84% | 73.91% | 54.14% | 52.03% |
| UDA [53] | 87.99% | 77.12% | 75.24% | 60.10% | 54.88% | 49.95% | 74.25% | 57.92% | 56.82% |
| 3DIoUMatch [11] | 87.86% | 77.26% | 75.28% | **63.17%** | 56.42% | 50.03% | 74.19% | 55.20% | 52.72% |
| UDA+CL [54] | 88.13% | 77.46% | 75.63% | 60.49% | 55.59% | 50.02% | 77.84% | 58.42% | 56.91% |
| SPSL-3D | **88.30%** | 77.81% | 75.58% | 62.84% | 56.97% | 54.59% | 77.90% | 58.13% | 57.02% |
| PSPSL-3D | 88.22% | **77.84%** | **75.76%** | 62.87% | **57.02%** | **54.88%** | **77.96%** | **58.21%** | **57.18%** |
| Gain from baseline | +1.02% | +0.70% | +0.32% | +3.74% | +4.47% | +4.39% | +4.11% | +4.00% | +4.92% |

**Table 5.** 3D AP of current SSOD-3D methods in KITTI validation dataset at $R_{\text{anno}} = 64\%$.

| 64% Labeled | 3D AP of Car | | | 3D AP of Pedestrian | | | 3D AP of Cyclist | | |
|---|---|---|---|---|---|---|---|---|---|
| Methods | Easy | Moderate | Hard | Easy | Moderate | Hard | Easy | Moderate | Hard |
| Baseline [21] | 88.94% | 78.70% | 77.84% | 64.95% | 60.43% | 56.46% | 77.20% | 58.13% | 57.04% |
| SESS [10] | 88.97% | 78.83% | 77.55% | 65.21% | 60.37% | 56.15% | 77.38% | 58.44% | 57.65% |
| UDA [53] | 88.74% | 78.72% | 77.34% | 64.31% | 61.19% | 56.02% | 77.52% | 57.93% | 57.34% |
| 3DIoUMatch [11] | 89.20% | 78.91% | 78.02% | 65.60% | 61.72% | 56.33% | 77.82% | 58.94% | **58.02%** |
| UDA+CL [54] | 88.84% | 78.74% | 77.85% | 64.64% | 61.35% | 56.48% | 77.73% | 58.25% | 57.62% |
| SPSL-3D | 89.08% | 78.69% | 77.98% | 65.27% | 62.02% | 56.52% | 76.27% | 58.80% | 56.44% |
| PSPSL-3D | **89.35%** | **78.99%** | **78.13%** | **65.75%** | **62.64%** | **56.72%** | **78.29%** | **60.71%** | 57.78% |
| Gain from baseline | +0.41% | +0.29% | +0.29% | +0.80% | +2.21% | +0.26% | +1.09% | +2.58% | +0.74% |

### 4.3. Comparison with Fully Supervised Methods

This experiment investigates the comparison results of the proposed PSPSL-3D methods with current FSOD-3D methods. FSOD-3D methods are all trained on the entire $\mathbb{D}_{\text{train}}$. For the proposed method PSPSL-3D at $R_{\text{anno}} = 100\%$, the unlabeled testing samples $\mathbb{D}_{\text{test}}$ in the KITTI dataset are used in the training procedure. Results in the KITTI validation dataset are provided in Table 6. The 3D APs of almost all categories of SPSL-3D and PSPSL-3D are smaller than fully supervised Voxel RCNN [21], but the 3D APs difference between car and pedestrian is not large. Compared with other FSOD-3D methods, it is found that 3D APs of pedestrians of PSPSL-3D are larger than some of classical methods [4,20,26], while 3D APs of the car category are smaller than the state-of-the-art FSOD-3D methods [14,29,55,56]. Additionally, the BEV AP results of SPSL-3D, PSPSL-3D, and fully supervised Voxel RCNN [21] are also presented in Table 7. As BEV object detection is easier than 3D object detection, it is found that most of the BEV APs of the proposed SSOD-3D methods and fully supervised methods are fairly close. In conclusion, the proposed PSPSL-3D method, which needs only 64% labeled data, can achieve a performance that is close to the BEV and 3D object detection performance of current FSOD-3D methods. By exploiting more unlabeled samples ($R_{\text{anno}} = 100\%$), the proposed PSPSL-3D outperforms than other FSOD-3D methods, which means that SSOD-3D has huge research potential in the field of autonomous driving.

**Table 6.** 3D AP of the proposed SSOD-3D at $R_{\text{anno}} = 64\%$ and current FSOD-3D methods in KITTI validation dataset.

| Methods | 3D AP of Car | | | 3D AP of Pedestrian | | | 3D AP of Cyclist | | |
|---|---|---|---|---|---|---|---|---|---|
| | Easy | Moderate | Hard | Easy | Moderate | Hard | Easy | Moderate | Hard |
| PointPillars [20] | 87.75% | 78.39% | 75.18% | 57.30% | 51.41% | 46.87% | 81.57% | 62.94% | 58.98% |
| Point-RCNN [4] | 88.26% | 77.73% | 76.67% | 65.62% | 58.57% | 51.48% | 82.76% | 62.83% | 59.62% |
| PV-RCNN [26] | 92.57% | 84.83% | 82.69% | 64.26% | 56.67% | 51.91% | 88.88% | 71.95% | 66.78% |
| 3D-CVF [14] | 89.97% | 79.88% | 78.47% | – | – | – | – | – | – |
| SE-SSD [29] | 93.19% | 86.12% | 83.31% | – | – | – | – | – | – |
| EPNet [55] | 92.28% | 82.59% | 80.14% | – | – | – | – | – | – |
| TANet [56] | 88.21% | 77.85% | 75.62% | 70.80% | 63.45% | 58.22% | 85.98% | 64.95% | 60.40% |
| Voxel RCNN [21] | 89.17% | 79.25% | 78.33% | 66.43% | 62.59% | 57.14% | 83.02% | 63.87% | 57.62% |
| SPSL-3D (Our, 64%) | 89.08% | 78.69% | 77.98% | 65.27% | 62.02% | 56.52% | 76.27% | 58.80% | 56.44% |
| PSPSL-3D (Our, 64%) | 89.35% | 78.99% | 78.13% | 65.75% | 62.64% | 56.72% | 78.29% | 60.71% | 57.78% |
| PSPSL-3D (Our, 100%) | **89.27%** | **79.32%** | **78.58%** | **68.11%** | **65.51%** | **59.58%** | **83.29%** | **64.30%** | **58.13%** |

**Table 7.** BEV AP of the proposed SSOD-3D at $R_{\text{anno}} = 64\%$ and baseline FSOD-3D methods in KITTI validation dataset.

| | BEV AP of Car | | | BEV AP of Pedestrian | | | BEV AP of Cyclist | | |
|---|---|---|---|---|---|---|---|---|---|
| Methods | Easy | Moderate | Hard | Easy | Moderate | Hard | Easy | Moderate | Hard |
| Voxel RCNN [21] | 90.17% | 88.17% | 87.41% | 67.86% | 63.28% | 59.97% | 82.87% | 64.17% | 57.85% |
| SPSL-3D (Our, 64%) | 90.12% | 87.84% | 87.04% | 69.23% | 63.34% | 58.14% | 76.89% | 64.33% | 56.84% |
| PSPSL-3D (Our, 64%) | 90.13% | 88.03% | 87.12% | 70.58% | 64.73% | 58.92% | 77.22% | 64.74% | 57.89% |
| PSPSL-3D (Our, 100%) | **90.40**% | **88.30**% | **87.86**% | **73.97**% | **67.83**% | **61.45**% | **83.38**% | **64.51**% | **58.30**% |

*4.4. Visualizations*

To better show the learning efficiency of PSPSL-3D, visualizations of 3D object detection are provided and discussed in this experiment. Comparisons with a fully supervised baseline 3D detector and PSPSL-3D trained in $R_{\text{anno}} = 64\%$-labeled data are shown in Figure 3. In the different outdoor scenarios, the proposed PSPSL-3D has nearly the same performance as the fully supervised baseline detector. If the object is far from the LiDAR, only a few false-positives are generated in PSPSL-3D. To further compare 3D object detection results in the LiDAR point cloud, more visualizations of the proposed method in the condition of $R_{\text{anno}=8\%}$ are presented in Figures 4 and 5. As 3DIoUMatch [11] has a performance close to that of PSPSL-3D, it is used for the main comparison. In the complex street scenarios with lots of cars and cyclists, PSPSL-3D has a higher 3D recall rate than 3DIoUMatch, because the proposed PSPSL-3D emphasizes the reliability of the pseudo label, thus achieving stable learning results. PSPSL-3D also has accurate results in simple outdoor scenarios. It is concluded that the proposed PSPSL-3D has stable and accurate 3D object detection performance.

| Fully supervised Baseline | PSPSL-3D (Our, 64%) |

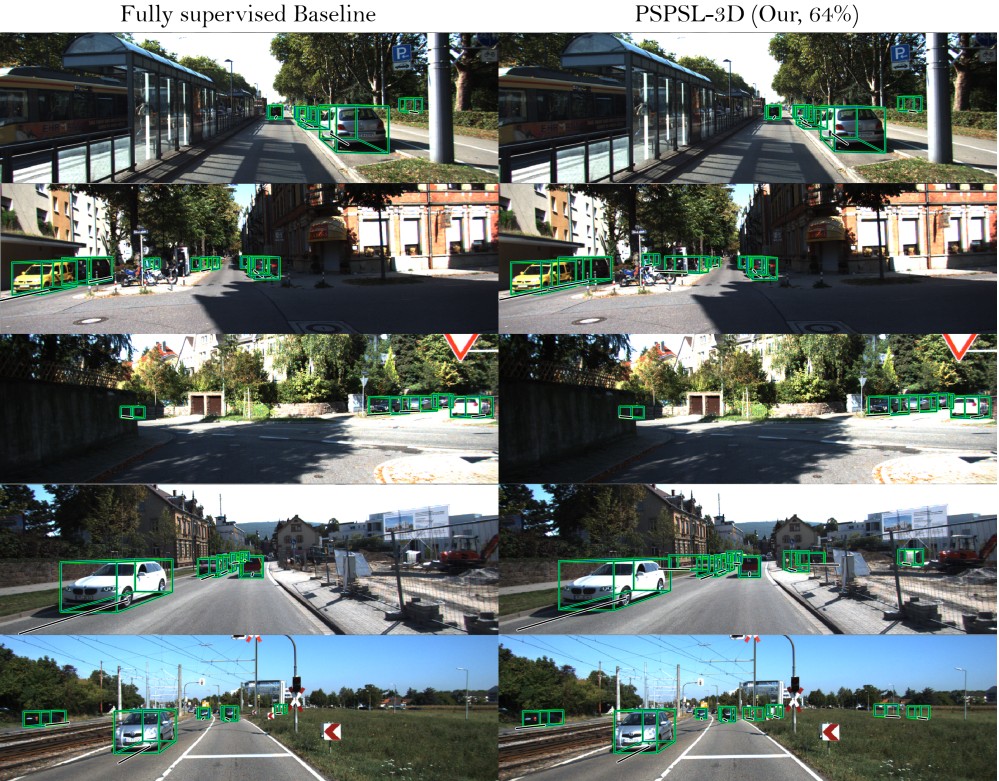

**Figure 3.** 3D object detection results of the fully supervised (FS) baseline method and the proposed PSPSL-3D trained in $R_{\text{anno}} = 64\%$-labeled data in the KITTI validation dataset.

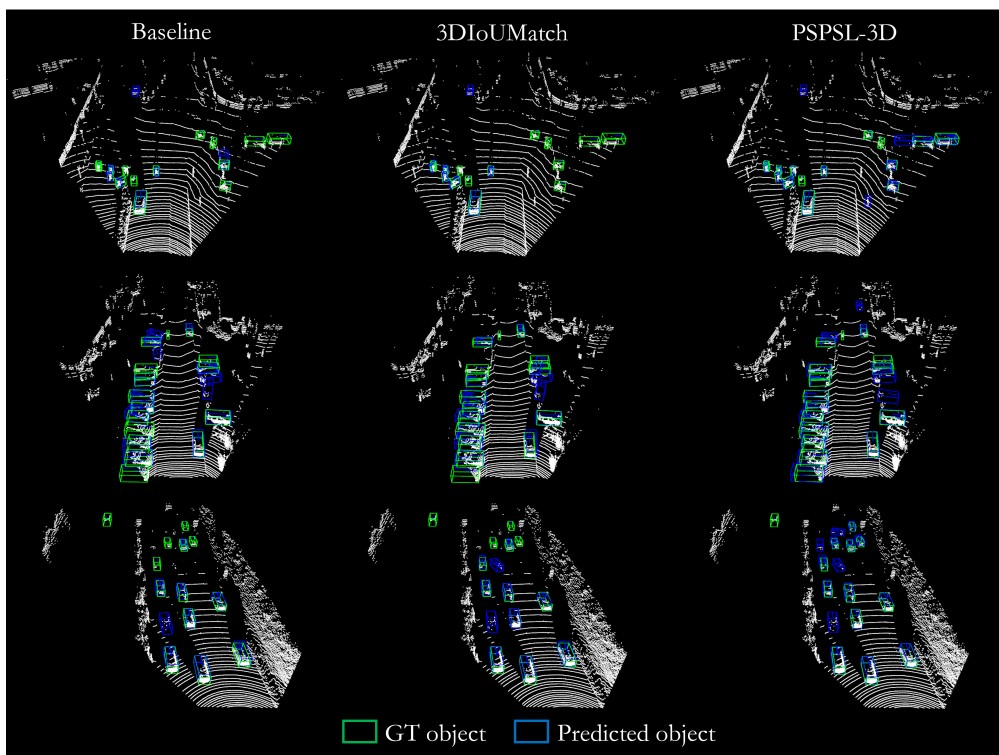

**Figure 4.** 3D object detection results in the complex scenarios. Baseline, 3DIoUMatch, and PSPSL-3D are trained with $R_{\mathrm{anno}} = 8\%$-labeled data. The recall rate of PSPSL-3D is higher than in other methods.

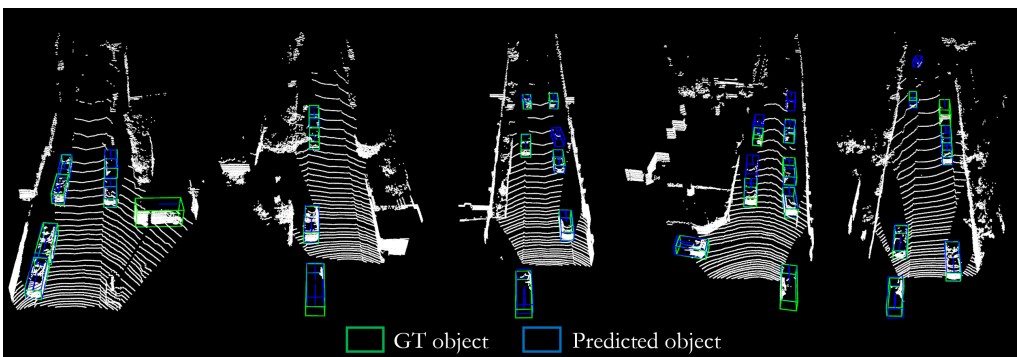

**Figure 5.** 3D object detection results of PSPSL-3D trained with $R_{\mathrm{anno}} = 8\%$-labeled data in simple traffic scenarios. Few false-positives and true-negatives were obtained.

### 4.5. Ablation Study

This experiment evaluates the effectiveness of the proposed PSPSL-3D framework. From Tables 6–8, it is also found that the 3D AP, BEV AP, and 3D recall rates of PSPSL-3D are higher than those of SPSL-3D in most of cases. Specifically, in the task of BEV object detection, PSPSL-3D has significantly improved pedestrian and cyclist detection under the same annotation conditions. In the task of 3D object detection, under the condition that $R_{\mathrm{anno}} \leq 16\%$, PSPSL-3D has a larger improvement than SPSL-3D in detecting 3D objects of all categories. The reason for this is provided in the following. Based on an SPSL-3D framework, PSPSL-3D exploits prior knowledge from the RGB image and LiDAR point cloud and then establishes extra regularization terms to prevent incorrectly predicted labels, thus achieving a higher 3D object detection performance. Therefore, ablation studies demonstrate the effectiveness of the proposed PSPSL-3D.

**Table 8.** 3D recall rate with an IoU threshold of 0.7 in the KITTI validation dataset.

| Method/$R_{anno}$ | 4% | 8% | 16% | 32% | 64% |
|---|---|---|---|---|---|
| SPSL-3D | 43.47% | 57.46% | 65.65% | 67.06% | 71.30% |
| PSPSL-3D | **44.25**% | **58.03**% | **66.13**% | **67.37**% | **71.63**% |

## 5. Discussion

The proposed SSOD-3D frameworks, SPSL-3D and PSPSL-3D, have several advantages. At first, we consider the reliability of pseudo label in the SSOD-3D training stage. As the GT annotation of pseudo label is unknown, we attempt to use the consistency loss to represent the weight of the pseudo label. If the pseudo label is incorrect, its consistency loss is larger than other pseudo labels. To reduce the negative effect of the pseudo label with large noise, we need to decrease the reliability weight of this pseudo label in the training stage. To adaptively and dynamically adjust the reliability weight of all pseudo labels, we exploit the theory of SPL [18] and then propose SPSL-3D as a novel and efficient framework. Second, we utilize the multi-model sensor data in the semi-supervised learning stage, thus further enhancing the capacity of the baseline 3D object detector based on LiDAR. The reason for the usage of multi-model data is that we notice that the LiDAR-camera system is widely equipped in the autonomous driving system. Thus, both the RGB image and LiDAR point cloud can be used in the training stage of SSOD-3D. For one object, there are generally abundant structural and textural features in the RGB image and LiDAR point cloud. However, as shown in Figure 6, the RGB image has shadow, occlusion, and blur, so it is difficult to extract prior knowledge of an object in the RGB image. Compared with RGB images, semantic segmentation images can directly reflect the object category information. Based on this analysis, we used the area of semantic segmentation of the object to describe the annotation accuracy. However, the semantic segmentation image also has two main limitations: it cannot identify occluded objects, and finds it hard to determine objects that are far from the sensor. Thus, the proposed PSPSL-3D exploits Equation (9) to approximately represent the annotation accuracy of the pseudo label.

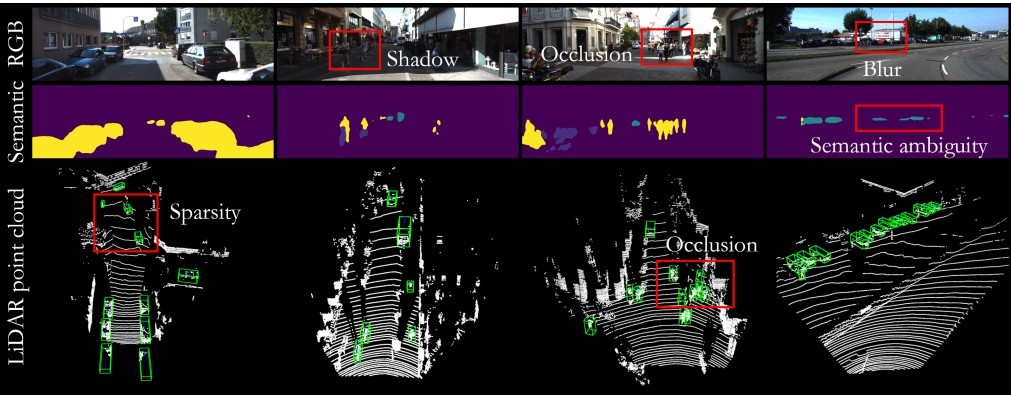

**Figure 6.** Multi-sensor data collected by the LiDAR-camera system in the different outdoor scenarios. Occlusion, shadow, blur, and sparsity in the LiDAR point cloud and RGB image have impacts on the detection difficulty and annotation accuracy (only for pseudo label) of the 3D object, limiting the efficiency of SSOD-3D. Blur in RGB images causes ambiguity in the semantic image. The image and 3D point cloud show significant differences in the various scenes. 3D objects are detected using the proposed PSPSL-3D framework with $R_{anno} = 100\%$.

Extensive experiments in Section 4 demonstrate the effectiveness of the proposed SPSL-3D and PSPSL-3D. Firstly, compared with the state-of-the-art SSOD-3D methods, the proposed SPSL-3D and PSPSL-3D frameworks have achieved the better results than other methods because the proposed frameworks consider the reliability of the pseudo labels, thus decreasing the negative effect of incorrect pseudo labels in the training stage. Second, the proposed frameworks are also suitable to train baseline 3D object in a fully

supervising way. Compared with the current FSOD-3D methods, the baseline 3D detector trained with PSPSL-3D outperforms other FSOD-3D methods. This means that the training scheme which emphasizes the weight of label is beneficial to baseline 3D object detector training.

In the future, we will study SSOD-3D in the several ways. Firstly, computer graphic (CG) techniques can be used to generate a huge number of labeled simulated samples. Exploiting the theory of SPL [57] in SSOD-3D with unlabeled samples and labeled simulated samples is an ongoing problem. Secondly, in actual application, the data distribution of the labeled LiDAR point cloud might be different from that of the unlabeled LiDAR point cloud because the dataset is collected with a different type of LiDAR sensor at a different place. Utilizing domain adaptation in a SSOD-3D is a challenging problem. We will deal with the above problems in subsequent studies.

## 6. Conclusions

The main challenge of learning-based 3D object detection is the shortage of labeled samples. To make full use of unlabeled samples, SSOD-3D is an important technique. In this paper, we propose a novel and efficient SSOD-3D framework for 3D object detection on a LiDAR-camera system. Firstly, to avoid the negative effect of unreliable pseudo labels, we propose SPSL-3D to adaptively evaluate the reliability weight of pseudo labels. Secondly, to better evaluate the reliability weight of pseudo labels, we utilize prior knowledge from the LiDAR-camera system and present the PSPSL-3D framework. Extensive experiments show the effectiveness of the proposed SPSL-3D and PSPSL-3D on the public dataset. Hence, we believe that the proposed framework benefits 3D environmental perception in autonomous driving.

**Author Contributions:** Methodology and writing—original draft preparation, P.A.; software, J.L.; investigation, X.H.; formal analysis, T.M.; validation, S.Q.; supervision, Y.C. and L.W.; Resources, and funding acquisition, J.M. All authors have read and agreed to the published version of the manuscript.

**Funding:** This research was funded by the National Natural Science Foundation of China (U1913602, 61991412, 62201536). Equipment Pre-Research Project (41415020202, 41415020404, 305050203).

**Data Availability Statement:** Publicly available datasets were analyzed in this study. This data can be found here: http://www.cvlibs.net/datasets/kitti/ (accessed on 1 March 2022).

**Acknowledgments:** The authors thank Siying Ke, Bin Fang, Junfeng Ding, Zaipeng Duan, and Zhenbiao Tan from Huazhong University of Science and Technology, and anonymous reviewers for providing many valuable suggestions.

**Conflicts of Interest:** The authors declare no conflict of interest.

## Abbreviations

The following abbreviations are used in this manuscript:

| | |
|---|---|
| LiDAR | Light detection and ranging |
| 3D | Three dimensional |
| SSL | Semi-supervised learning |
| SSOD-3D | Semi-supervised 3D object detection |
| FSOD-3D | Fully supervised 3D object detection |
| WSOD-3D | Weakly supervised 3D object detection |
| SPL | Self-paced learning |
| SPSL-3D | Self-paced semi-supervised learning based 3D object detection |
| PSPSL-3D | Self-paced semi-supervised learning based 3D object detection with prior knowledge |
| AP | Average precision |
| BEV | Bird's eye view |
| UDA | Unsupervised data augmentation |
| CL | Curriculum learning |

| | |
|---|---|
| IoU | Intersection over union |
| GT | Ground truth |
| EMA | Exponential moving average |
| NMS | Non maximum suppression |
| FC | Fully connected |
| CNN | Convolutional neural network |
| GNN | Graph neural network |
| Spconv | Sparse convolutional neural network |

## Appendix A

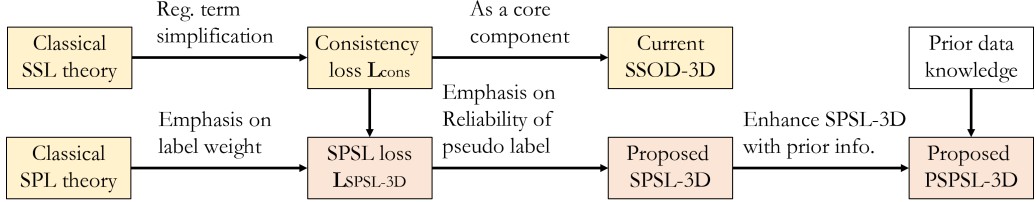

**Figure A1.** Relation of classical SSL, current SSOD-3D, and proposed SPSL-3D, PSPSL-3D. As weight function $w(\mathbf{x}_i, \mathbf{x}_j)$ is difficult to design in complex traffic scenarios, the regularization (reg.) term $\|\mathbf{w}\|_I$ in traditional SSL theory is simplified as the consistency loss $\mathrm{L}_{\mathrm{cons}}$. In this paper, emphasising the impact of the reliability of pseudo labels on SSOD-3D training, we present SPL-based consistency loss $\mathcal{L}_{\mathrm{SPSL-3D}}$ and SPSL-3D. Prior information (info.) is exploited to enhance the effect of SPL on SPSL-3D.

*Appendix A.1. Relation of Traditional SSL Theory and Previous SSOD-3D Method*

We briefly revisit the previous SSOD-3D approach [10] and discuss the relation between the classical SSL theory and existing SSOD-3D method (seen in Figure A1). Classical SSL [37] aims to find $\mathbf{w}^*$ by minimizing the following cost function:

$$\mathcal{L}_{\mathrm{SSL}}(\mathbf{w}) = \sum_{i=1}^{\mathrm{N}_l} \|\mathrm{L}_{3d}(\mathbf{l}_i^p, \mathbf{l}_i)\|_1 + \alpha\|\mathbf{w}\|_2^2 + \gamma\|\mathbf{w}\|_I^2, \mathbf{x}_i \in \mathbb{X}_l \tag{A1}$$

where $\alpha$ and $\gamma$ are coefficients of regularization terms. Suppose that $\mathbf{x}$ lies in a compact manifold $\mathbb{M}$ [37]. $\mathcal{P}_\mathbf{x}$ is the marginal distribution of $\mathbf{x}$. $\|\mathbf{w}\|_I$ reflects the intrinsic structure of $\mathcal{P}_\mathbf{x}$ on $\mathbb{M}$ [37]:

$$\begin{aligned} \|\mathbf{w}\|_I^2 &= \int_{\mathbf{x}\in\mathbb{M}} \|\nabla_\mathbb{M}\mathbf{w}\|^2 d\mathcal{P}_\mathbf{x}(\mathbf{x}) \\ &\approx \frac{1}{\mathrm{N}_l + \mathrm{N}_u} \sum_{i\neq j} w(\mathbf{x}_i, \mathbf{x}_j) \mathrm{L}_{3d}(\mathbf{l}_i^p, \mathbf{l}_j^p), \mathbf{x}_i, \mathbf{x}_j \in \mathbb{X}_l \cup \mathbb{X}_u \end{aligned} \tag{A2}$$

In the above equation, data weight $w(\mathbf{x}_i, \mathbf{x}_j) \in [0,1]$ describes the similarity of $\mathbf{x}_i$ and $\mathbf{x}_j$. However, as presented in Figure 6, $\mathbf{x}_i$ and $\mathbf{x}_j$ collected in the different scenarios have different data distribution. Their object number and category are also not the same. Therefore, it is hard to design a suitable $w(\mathbf{x}_i, \mathbf{x}_j)$. There is a gap between traditional SSL theory and SSOD-3D.

To solve this problem, researchers attempted to relax $\|\mathbf{w}\|_I$ and present the consistency loss $\mathrm{L}_{\mathrm{cons}}$ as an approximation of SSOD-3D. They exploit data augmentation operator $\mathbf{A}(\cdot)$ to create datum $\mathbf{A}(\mathbf{x}_i)$ similar to $\mathbf{x}_i$ [10,39]. $\mathbf{A}(\mathbf{x}_i)$ is used to replace $\mathbf{x}_j$ in Equation (A2). $\mathbf{A}(\mathbf{x}_i)$ is the affine transformation on $\mathbf{P}_i$ of $\mathbf{x}_i$, which contains scaling, X/Y-axis flipping and Z-axis rotating operations [26]. As affine transformation does not change the structure of point cloud, it is safe to assume that $w(\mathbf{x}_i, \mathbf{A}(\mathbf{x}_i)) \approx 1$. $\mathbf{l}_{i,\mathrm{Aug}}^p$ is one of the results in $f(\mathbf{A}(\mathbf{x}_i); \mathbf{w})$. With the inverse affine transformation $\mathbf{A}^{-1}(\cdot)$, $\mathbf{A}^{-1}(\mathbf{l}_{i,\mathrm{Aug}}^p)$ is obtained, which has the same reference coordinate system as $\mathbf{l}_i^p$. After that, $\|\mathbf{w}\|_I^2$ in Eq. (A1) is simplified

and replaced as one consistency loss L$_{cons}$ [10] as Equation (2). It is the core loss function in the pseudo label based SSOD-3D methods [10,11,38,41,47]. In the other literature, L$_{cons}$ is called unsupervised data augmentation (UDA) [53] because it does not utilize any GT information. The current SSOD-3D [10] optimizes $\mathbf{w}^*$ by minimizing the function in Equation (1).

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
