# Peer review of "Leveraging Self-Paced Semi-Supervised Learning with Prior Knowledge for 3D Object Detection on a LiDAR-Camera System"

_remotesensing, doi:10.3390/rs15030627_

Round 1
Reviewer 1 Report
Dear authors,
Thank you for your efforts in producing this interesting paper.
The paper addresses a relevant topic of a semi-supervised learning technique for 3D object detection. The authors demonstrated, on several cases, the use of their algorithm in comparison with state-of-the-art algorithms.
I didn’t find any significant flaws in the article, i.e., in the methodology, presentation and results.
I have only a few comments below.
Line 221: Figure 1 should be placed when first mentioned.
Line 297: based on what were the values of Ranno set? Many readers may need clarification on why these values were chosen. Assumptions of the authors?
As I stated before, I found the paper generally well written; perhaps I would have welcomed more illustrations on figures regarding the proposed algorithm's results, comparison, and advantages.
Regards
Author Response
Comments:
Thank you for your efforts in producing this interesting paper.
The paper addresses a relevant topic of a semi-supervised learning technique for 3D object detection. The authors demonstrated, on several cases, the use of their algorithm in comparison with state-of-the-art algorithms.
[Authors] Thanks for the comments. We wish the proposed method benefits the community of LiDAR based 3D environmental perception in the field of remote sensing. We proofread the manuscript after revision, check and correct the grammar mistakes. Please refer the red-line version of the revised manuscript.
OC1:
I didn’t find any significant flaws in the article, i.e., in the methodology, presentation and results. I have only a few comments below.
Line 221: Figure 1 should be placed when first mentioned.
Line 297: based on what were the values of Ranno set?
Many readers may need clarification on why these values were chosen. Assumptions of the authors?
[Authors] Thanks for the suggestions. At first, we revise the sentences in Sec. 3.2 and 3.3 to make sure that Fig. 1 is placed when it is first mentioned. Second, we are sorry for the unclear illustration on the value setting of R_anno. It is noted that semi-supervised learning (SSL) is an algorithm to train model in both labeled and unlabeled datasets. We attempt to evaluate the proposed SSL with the various situations with different labeled ratios from 4% (hard case, lots of samples are unlabeled) to 64% (an easy case, lots of samples are labeled), to show readers that the proposed SPSL-3D is stable and efficient.
Sec. 3.2 Previous semi-supervised 3D object detection (Page 5)
The pipeline of the previous SSOD-3D is presented in Fig. 1(a).
Sec. 3.3 Self-paced semi-supervised learning based 3D object detection (Page 6)
Based on this fact, SPSL-3D is proposed, and its pipeline is presented in Fig. 1(b).
Sec. 3.3 Self-paced semi-supervised learning based 3D object detection (Page 6)
The improvement can be found in Fig. 1(c).
Sec. 4.1 Dataset and configuration (Page 9)
In the following experiments, to evaluate the performance of SSOD-3D comprehensively, we set the various training situations with the different labeled ratios, from 4% (Hard SSL case) to 64% (Easy SSL case). Specifically, Ranno is set as 4%, 8%, 16%, 32%, 64%, respectively. And we mainly focus on the comparison results in the hard SSL case (Ranno≤ 16%).
OC2:
As I stated before, I found the paper generally well written; perhaps I would have welcomed more illustrations on figures regarding the proposed algorithm's results, comparison, and advantages.
[Authors] Thanks for your warm comments. To make the readers easy to follow the idea of the proposed SSOD-3D, we polish the illustrations on Figs. 1 and 2 (about algorithms), Figs. 3, 4, 5 (about comparison results and advantages).
For your convenience, main revisions are presented in the following.
About algorithms (Figs. 1 and 2)
Sec. 3.2 Previous semi-supervised 3D object detection (Page 5)
The pipeline of the previous SSOD-3D is presented in Fig. 1(a). For the 3D object detector with the high generalization ability, its prediction results from the unlabeled sample xi and its augmented sample A(xi) are both consistent and closed to the GT labels. Based on this analysis, as the unlabeled sample does not have annotation, Lcons was proposed to minimize the difference of labels predicted from xi and A(xi).
Sec. 3.2 Previous semi-supervised 3D object detection (Page 5)
Lcons is the core in this scheme [10], for this loss can update the weights in 3D object detector via back-propagation.
Sec. 3.3 Self-paced semi-supervised learning based 3D object detection (Page 6)
One naive idea is to adjust the reliability weight vj with the guidance of the consistency loss of lp. If the consistency loss of lp enlarges, the pseudo labels lp are unreliable. Based on this idea, we exploit the theory of SPL [18] to construct the mathematical relation of vj and L3d, and propose a novel SSOD-3D framework SPSL-3D in this paper.
Sec. 3.4 Improving SPSL-3D with prior knowledge (Page 7)
In fact, the reliability weight of pseudo label is not only dependent on the consistency loss, but also dependent on the prior knowledge in LiDAR point cloud and RGB image provided by LiDAR-camera system. If LiDAR point cloud or image feature of one predicted object is not salient, its pseudo label is not reliable. Based on this analysis, to further enhance the performance of PSPL-3D with the information from LiDAR point cloud and RGB image, we propose a prior knowledge based SPSL-3D named as PSPSL-3D, which is presented in Fig. 2.
Sec. 3.4 Improving SPSL-3D with prior knowledge (Page 7)
In PSPSL-3D, we attempt to represent the reliability of pseudo label with LiDAR point cloud and RGB image.
About comparison results and advantages (Figs. 3, 4, and 5)
Sec. 4.4 Visualizations (Page 12)
In the different outdoor scenarios, the proposed PSPSL-3D has nearly the same performance as the fully supervised baseline detector. If the object is far from LiDAR, only few false-positives are generated in PSPSL-3D.
Sec. 4.4 Visualizations (Page 12)
In the complex street scenarios with lots of cars and cyclists, PSPSL-3D has the higher 3D recall rate than 3DIoUMatch, for the proposed PSPSL-3D emphasizes on the reliability of pseudo label, thus achieving the stable learning results.

Reviewer 2 Report
The authors presents a novelty self-paced semi-supervised learning with prior knowledge for 3D object detection using a LiDAR-camera system.
The paper is well written and the organized. The results are adequately presented. I consider that the experiments are sufficient to demonstrate the effectiveness of such a scheme.
Author Response
Comments:
The authors presented a novelty self-paced semi-supervised learning with prior knowledge for 3D object detection using a LiDAR-camera system.
The paper is well written and the organized. The results are adequately presented. I consider that the experiments are sufficient to demonstrate the effectiveness of such a scheme.
[Authors] Thanks for your comments. We think that it is meaningful to exploit the theory of self-paced learning into semi-supervised 3D object detection. In this work, we study the approach to refine self-paced learning with considering the prior knowledge of sensor data, which benefits the researchers in the field of autonomous driving.
To better improve the quality of manuscript, we carefully proofread the manuscript, and revise several grammar mistakes, and lots of sentences to make the idea of SPSL-3D clearly and easy to follow. Please refer the red-line version of the revised manuscript.

Reviewer 3 Report
The manuscript is well organized and neatly written.
To me, the key contribution of the current work is the introduction and exploitation of the traffic scene priors in the form of regularization on the weighting vector v in Eq(10). Concerning the so-called self-paced training, it is merely an adaptation of a training procedure proposed in [NIPS2010], it can hardly constitute any significant contribution per se.
I have two major concerns over the manuscript:
First, from the reported comparison experiments, it seems the performance ( Table 1 - Table 5) of the proposed SPSL-3D, PSPSL-3D is comparable with 3DIoUMatch, no noticeable improvement is obtained. In these tables, the reported GAIN ( in red) is somehow misleading. This gain is over Baseline, not over the best one of the comparison methods;
From the ablation study, it seems PSPSL-3D performs only slightly better than SPSL-3D, this means the key contribution of this work, i.e., the exploitation of traffic scene knowledge in Eq(10), seems to play only a minor role.
My second concern is the suitability of the topic to this journal. My understanding is that Remote Sensing should cover the processing of remotely sensed data, this work is purely a computer vision topic. It seems no good for RS to be stretched to natural images.
Some typos:
Line 51: IoU: Intersection ( not Interaction) over UnionLine 162: Time-efficient ( not time-efficiency)
Line 213: noised->noisy
Line 249: is mainly depended on ( dependent on)
Line 286: KITTI dataset is exploited ( used. Exploitation means you make efforts to do sth)
All in all, considering the slight performance improvement and topic fitness issue, I would give this manuscript a “borderline” mark. If the editorial board thinks the topic is appropriate and no more high quality submissions are available, it could be accepted for publication, otherwise it could be rejected.
Author Response
Comments:
The manuscript is well organized and neatly written. To me, the key contribution of the current work is the introduction and exploitation of the traffic scene priors in the form of regularization on the weighting vector v in Eq. (10). Concerning the so-called self-paced training, it is merely an adaptation of a training procedure proposed in [NIPS2010], it can hardly constitute any significant contribution per se.
[Authors] Thanks for your comments. Actually, we have the original idea when we exploits the theory of self-paced learning into semi-supervised 3D object detection. Thus, we wish to clarify the contributions in this manuscript.
- about SPSL-3D. The main problem of SSOD-3D is the inaccuracy of pseudo label. To decrease the negative effect of incorrect pseudo label, we have to evaluate the reliability of each pseudo label. In this way, the next problem is to measure the reliability of each pseudo label. SPL has the advantage of modeling the weight of sample with its loss. So, we exploit SPL into SSOD-3D. After that, we have to consider which self-paced regularization term is used in SSOD-3D (discussion is presented in line 239). Also, we have to consider how to design the age parameter in SPL. In a word, when we use SPL in SSOD-3D, we discuss the in-depth reason why use SPL, and provide the technique details how to use SPL. These details benefit the researcher who is interesting to both SPL and SSOD-3D. Therefore, we consider that SPSL-3D is one of the core contributions in this paper.
- about PSPSL-3D. In the classic SPL theory, self-paced regularization term is dependent on the learning loss. However, when SPL is exploited in one concrete application (i.e. 3D object detection), it is essential to study the weight of sample from the viewpoint of sensor data (i.e. point cloud and RGB image). If the object feature is not salient in the sensor data, its pseudo label is not reliable enough. Based on this analysis, we design a novel self-paced regularization term based on the prior knowledge of LiDAR point cloud and RGB image. Therefore, we consider that PSPSL-3D is another core contributions in this paper.
From the above discussions, we believe that the novelty of the proposed SSOD-3D method is relatively significant, and our work benefits the community of the advanced vision task in remote sensing.
Sec. 3.3 Self-paced semi-supervised learning based 3D object detection (Page 6)
One naive idea is to adjust the reliability weight vj with the guidance of the consistency loss of lp. If the consistency loss of lp enlarges, the pseudo labels lp are unreliable. Based on this idea, we exploit the theory of SPL [18] to construct the mathematical relation of vj and L3d, and propose a novel SSOD-3D framework SPSL-3D in this paper.
Sec. 3.4 Improving SPSL-3D with prior knowledge (Page 7)
In fact, the reliability weight of pseudo label is not only dependent on the consistency loss, but also dependent on the prior knowledge in LiDAR point cloud and RGB image provided by LiDAR-camera system. If LiDAR point cloud or image feature of one predicted object is not salient, its pseudo label is not reliable. Based on this analysis, to further enhance the performance of PSPL-3D with the information from LiDAR point cloud and RGB image, we propose a prior knowledge based SPSL-3D named as PSPSL-3D, which is presented in Fig. 2.
OC1:
First, from the reported comparison experiments, it seems the performance (Table 1 - Table 5) of the proposed SPSL-3D, PSPSL-3D is comparable with 3DIoUMatch, no noticeable improvement is obtained. In these tables, the reported GAIN (in red) is somehow misleading. This gain is over Baseline, not over the best one of the comparison methods;
From the ablation study, it seems PSPSL-3D performs only slightly better than SPSL-3D, this means the key contribution of this work, i.e., the exploitation of traffic scene knowledge in Eq. (10), seems to play only a minor role.
[Authors] Thanks for your comments. We are sorry for the misunderstanding of GAIN in Table 1 to 5. “Gain” is revised as “Gain from baseline”. As 3DIoUMatch is a well-designed method for SSOD-3D, it is not an easy task to design a novel SSOD-3D method which can have very large improvement on 3DIoUMatch. Well, from Table 1 to 5, improvements can be found from PSPSL-3D where the gain of cyclist is large significantly. It shows that the proposed PSPSL-3D has the better performance than 3DIouMatch. Moreover, it is the first time to exploit SPL theory into SSOD-3D, which is an interesting topic in the community of autonomous driving. In ablation study, compared with SPSL-3D, the gain of PSPSL-3D is larger significantly if Ranno is smaller than 16%. We will refine the experiment analysis in the revised manuscript.
Please refer the red-version manuscript. Main revisions are presented in the following.
Table 1 (Page 10)
please refer the Word file for the better visulization.
Sec. 4.2 Comparison with semi-supervised methods (Page 11)
It is found that 3DIouMatch [11] has the better performance than other previous methods. Compared with 3DIoUMatch, the proposed PSPSL-3D has the significant improvement on 3D cyclist detection, and also has the certain gain on 3D pedestrian detection, for the prior knowledge from LiDAR point cloud and RGB image is beneficial to modeling the reliability of object with the relatively small size.
Sec. 4.5 Ablation studies (Page 13)
Specifically, in the task of BEV object detection, PSPSL-3D has the significant improvement on pedestrian and cyclist detection under the same annotation condition. In the task of 3D object detection, under the condition that Ranno ≤ 16%, PSPSL-3D has the larger gain than SPSL-3D on detecting 3D objects of all categories.
OC2:
My second concern is the suitability of the topic to this journal. My understanding is that Remote Sensing should cover the processing of remotely sensed data, this work is purely a computer vision topic. It seems no good for RS to be stretched to natural images.
[Authors] Thanks for your comments. We consider that our work is not purely a computer vision topic. It has relation with processing remote sensing data. LiDAR-camera system is the common sensor in the field of remote sensing. In the proposed method, we design the regularization term based on the prior knowledge of LiDAR point cloud and RGB image. And, the task of 3D object detection on LiDAR-camera system is also common in the field of remote sensing.
Moreover, there are some newly published literatures in Remote Sensing:
(i) Mingming Wang, Qingkui Chen, Zhibing Fu.
LSNet: Learned Sampling Network for 3D Object Detection from Point Clouds. Remote. Sens. 14(7): 1539 (2022);
(ii) Feng Shuang, Hanzhang Huang, Yong Li, Rui Qu, Pei Li.
AFE-RCNN: Adaptive Feature Enhancement RCNN for 3D Object Detection. Remote. Sens. 14(5): 1176 (2022);
(iii) Ling Bai, Yinguo Li, Ming Cen, Fangchao Hu.
3D Instance Segmentation and Object Detection Framework Based on the Fusion of Lidar Remote Sensing and Optical Image Sensing. Remote. Sens. 13(16): 3288 (2021);
Based on the above discussion, we believe that the proposed method is suitable to Remote Sensing.
OC3:
Some typos:
Line 51: IoU: Intersection (not Interaction) over Union
Line 162: Time-efficient (not time-efficiency)
Line 213: noised->noisy
Line 249: is mainly depended on (dependent on)
Line 286: KITTI dataset is exploited (used. Exploitation means you make efforts to do sth)
[Authors] Thanks for your comments. We are sorry for these mistakes.
After the revision, we have proofread the manuscript carefully, and try our best to revise the grammar mistakes and incorrect expressions. Please refer the red-version manuscript in the submission.
